# Identifying Potential Super-Spreaders and Disease Transmission Hotspots Using White-Tailed Deer Scraping Networks

**DOI:** 10.3390/ani13071171

**Published:** 2023-03-26

**Authors:** Scoty Hearst, Miranda Huang, Bryant Johnson, Elijah Rummells

**Affiliations:** 1The Department of Chemistry and Biochemistry, Mississippi College, Clinton, MS 39056, USA; 2Department of Wildlife, Fisheries, and Aquaculture, Mississippi State University, Starkville, MS 39762, USA

**Keywords:** *Odocoileus virginianus*, white-tailed deer, scraping networks, social networks, potential super-spreaders, disease transmission hotspot, community crossroads, disease management and prevention strategies, predator activity, hunter harvests

## Abstract

**Simple Summary:**

White-tailed deer (WTD) spread communicable diseases such as the coronavirus SARS-CoV-2, which is a major public health concern, and chronic wasting disease (CWD), a fatal, highly contagious, brain disease occurring in species of the deer family. Currently, it is not well understood how WTD are spreading these diseases. In this paper, we speculate that “super-spreaders” mediate disease transmission via direct social interactions and indirectly via body fluids exchanged at scrape sites. Super-spreaders are infected individuals that infect more contacts than other infectious individuals within a population. Using social network analysis, we identified potential super-spreaders among multiple communities of a rural WTD herd in Mississippi. Analysis of age structure revealed that the majority of potential super-spreaders were young males, less than 2.5 years of age. We also compared infection risk across the landscape by combining social network analysis and heatmapping software to locate disease transmission hotspots, where the risk of disease transmission is higher as compared to other locations. We also monitored predator and hunting activity and hunter deer harvests to better understand how predators influence social networks and potential disease transmission. We found that predator activity influenced the age structure of male WTD communities. We assessed disease-management strategies by social-network modeling using hunter harvests or the removal of potential super-spreaders, which fragmented WTD social networks reducing the potential spread of disease. Overall, this study demonstrates a model for predicting potential super-spreaders of diseases, describes new methods to locate transmission hotspots, and provides new knowledge for disease management and prevention strategies.

**Abstract:**

White-tailed deer (*Odocoileus virginianus*, WTD) spread communicable diseases such the zoonotic coronavirus SARS-CoV-2, which is a major public health concern, and chronic wasting disease (CWD), a fatal, highly contagious prion disease occurring in cervids. Currently, it is not well understood how WTD are spreading these diseases. In this paper, we speculate that “super-spreaders” mediate disease transmission via direct social interactions and indirectly via body fluids exchanged at scrape sites. Super-spreaders are infected individuals that infect more contacts than other infectious individuals within a population. In this study, we used network analysis from scrape visitation data to identify potential super-spreaders among multiple communities of a rural WTD herd. We combined local network communities to form a large region-wide social network consisting of 96 male WTD. Analysis of WTD bachelor groups and random network modeling demonstrated that scraping networks depict real social networks, allowing detection of direct and indirect contacts, which could spread diseases. Using this regional network, we model three major types of potential super-spreaders of communicable disease: in-degree, out-degree, and betweenness potential super-spreaders. We found out-degree and betweenness potential super-spreaders to be critical for disease transmission across multiple communities. Analysis of age structure revealed that potential super-spreaders were mostly young males, less than 2.5 years of age. We also used social network analysis to measure the outbreak potential across the landscape using a new technique to locate disease transmission hotspots. To model indirect transmission risk, we developed the first scrape-to-scrape network model demonstrating connectivity of scrape sites. Comparing scrape betweenness scores allowed us to locate high-risk transmission crossroads between communities. We also monitored predator activity, hunting activity, and hunter harvests to better understand how predation influences social networks and potential disease transmission. We found that predator activity significantly influenced the age structure of scraping communities. We assessed disease-management strategies by social-network modeling using hunter harvests or removal of potential super-spreaders, which fragmented WTD social networks reducing the potential spread of disease. Overall, this study demonstrates a model capable of predicting potential super-spreaders of diseases, outlines methods to locate transmission hotspots and community crossroads, and provides new insight for disease management and outbreak prevention strategies.

## 1. Introduction

White-tailed deer (*Odocoileus virginianus*, WTD) display a seasonal social behavior called scraping, where physical and chemical signposts are used to advertise their sociosexual status [1]. WTD scraping behavior is often displayed as a series of behaviors as follows: sniffing an over-hanging branch (the licking branch); licking or tasting the branch; rubbing the branch with their antlers, pre-orbital gland or forehead; disturbing the soil below the branch with their hooves; and urinating on the disturbed soil [1,2,3,4,5]. Scraping is a scent marking behavior, where WTD deposit semiochemicals from a variety of different scent glands including: salivary, pre-orbital, forehead, interdigital, tarsal, and metatarsal glands [1,4,6]. The majority of WTD scrapes are made by mature males (≥2.5 years old) beginning near the onset of breeding season as a sociosexual communication displaying dominance to suppress competing males and to show prowess to potential mates [2,3,5,7,8]. Scrapes help establish male WTD social structure, where social rank, age, experience, and testosterone levels are major influencers of scraping behavior [2,8]. Female WTD also scrape, but not as often as males [1,2,8]. WTD scraping behaviors are olfactory communications used to form social networks during the breeding season [1,2,3,4,6].

Previously, we demonstrated the first method of using scraping behavior to generate male WTD social networks for disease modeling [2]. Social-network modeling can predict disease transmission, measure outbreak potential, and identify potential super-spreaders [2,9,10,11]. Super-spreaders are infected individuals that are highly connected within a social network and infect more social contacts than other infectious individuals within a population [2,9,10,11]. We speculated that WTD social-network models can be used to identify potential super-spreaders within WTD communities and be used to locate disease transmission hotspots leading to more effective disease-management strategies. 

Two major diseases of concern impacting WTD populations are chronic wasting disease (CWD), a fatal prion disease occurring in cervids, and the zoonotic SARS-CoV-2 virus, which is a major public health concern [12,13]. Presently, it is not well understood how WTD are spreading these communicable diseases and the how the outbreak potential varies over different locations. CWD prions have been detected in multiple WTD scent glands and secretions used in scraping behavior [13]. Since both prions and SARS-CoV-2 have been detected in the nasal and oral secretions of WTD, research points to scrapes as potential indirect transmission routes for these pathogens [14,15,16].

The goal of this study was to use scraping behavior to model disease transmission in a large WTD social network containing multiple communities, to identify potential super-spreaders, and to locate potential disease transmission hotspots and community crossroads. Overall, this study demonstrates the latest method capable of predicting potential super-spreaders of diseases among WTD populations, outlines a new technique to locate transmission hotspots and community crossroads, and provides new insight for WTD disease management and outbreak prevention strategies.

## 2. Methods

### 2.1. Study Site Description

Our study area was a 9.7 km × 0.8 km plot of private land, managed by multiple landowners, located in Bentonia, MS, USA. This location included open pastures, farmland, food-plots, acorn-producing hardwood forests, and mixed hardwood–pine forests; major farm crops at this location are cotton, corn, and soybeans. Over the survey period, we monitored 33 WTD scrapes spread over 3 major study sites and 3 boundary sites located at various points along the perimeter of study area (Figure 1). These study sites distanced 1.5 to 4.5 km apart with the goal of detecting multiple WTD communities within one region-wide social network. Major Site 1, Major Site 2, Boundary Site 1, and Boundary Site 2 were located near open farmland with sparse forest cover. Major Site 3 and Boundary Site 3 were located within dense forested regions where open fields were sparse. WTD are the only cervid species at the survey location. Coyotes (*Canis latrans*) and bobcats (*Lynx rufus*) are the dominant predator species in the area. WTD hunting was prevalent throughout the survey area as well as in the areas surrounding our survey location. Local landowners were surveyed to determine the number of hunters within and near our survey location. Hunters were questioned about male WTD harvested by both adult and youth hunters using protocol #11162021 approved by the Mississippi College IRB committee. Hunters were asked to share images of male WTD harvests. Hunter-harvests were recorded and matched to male WTD cataloged in this study.

### 2.2. Scrape Monitoring

To survey the scraping behavior, we used 33 camera traps, (Model #: 119270CW, Tasco Worldwide, Miami, FL, USA) positioned over 33 active scrapes and set in 3/5 mode (3 digital images recorded every 5 s of camera activation) using methods as previously described [2]. Briefly, we identified scrape sites by large circular depressions of disturbed ground under low-lying tree limbs using methods as previously described [2]. We deployed camera traps as scrapes began to appear (5 November 2020) and removed them 31 January 2021. The cameras were undisturbed except for memory card and battery changes, which we performed at 30-day intervals. Scraping behavior was defined as any marking behavior that might leave behind a scent communication such as: pawing the ground, marking overhead branches, and urination; while visiting behavior was defined as pausing at the scrape to investigate without performing any scraping behavior [2,3]. Scraping activity was measured for each scrape and defined as the number of scraping behaviors recorded at that scrape over the survey period. Unique male WTD were identified and cataloged as previously described [2,17]. A catalog of male WTD for each study site and boundary site was produced and compared to remove any duplicates over the multiple study sites. Unique profiles of each male were created using parameters such as: antler patterns, pelage, body size, and maturity estimation; maturity estimation was based upon body size and neck girth defining mature males as >2.5 years of age and young males as ≤2.5 years of age following methods as previously described [2,3,17,18]. Bachelor grouping behavior was defined as two or more male WTD seen traveling together on multiple occasions. Sparring/fight behavior was defined as physical antler contact made between two male WTD. Camera trap data was also used to monitor average predator activity per scrape at each study site. Since hunting is a predatory behavior and hunting pressure may influence WTD behavior, we defined predator activity as the number of hunters, coyotes, and stray dogs captured on film near a scrape site per week. Average predator activity was statistically compared between major study sites 1, 2 and 3. The percentage of mature and young males in the population scraping at each scrape site was averaged by study site location and was statistically compared. Statistical analysis was performed using a one-way ANOVA test, where *p* < 0.05 was considered significant. All distributions were tested using the Jarque–Bera test [19] with a threshold of *p* = 0.05 and appeared to be normally distributed.

### 2.3. Social Network Analysis

For the male WTD scraping networks, we recorded the number of scraping behaviors, visiting behaviors, and the scrape site locations for each male over all major and boundary study sites using methods as previously described [2]. Scraping data was interpreted as directional weighted edges targeted to each of the other males (nodes) that visited or scraped at the scrape site. We generated a male scraping network for each study site and boundary location using Gephi software [2,20]. These site-specific networks were also merged into one large Regional Network. Network data with relevance to disease transmission (Table 1) were calculated using Gephi software or using methods as previously described [2,20,21,22,23]. We used ANOVA tests to compare network data between study sites and between males to evaluate WTD potential super-spreaders. Closeness scores between male WTD implies direct contact where disease transmission can occur [2]. To explore closeness scores, we generated a Random Network by randomizing the edge weights and target nodes from the Regional Network data. Random networks are used for hypothesis testing as comparisons to real networks [24]. Closeness scores from the Regional Network and the Random Network were compared for bachelor groups and incidents of sparring/fighting behavior using the ANOVA test, where *p* < 0.05 was considered significant. All distributions were tested using the Jarque–Bera test [19] with a threshold of *p* = 0.05 and appeared to be normally distributed. Since out-degree centrality, in-degree centrality, and betweenness centrality scores can be used to identify potential super-spreaders within social networks [2,9,10,11], we ranked individual male WTD based on these variables using the Regional Network to identify potential super-spreaders using both an alpha of α = 1.0 and an α = 0.5 [2,21]. Alpha is used to modulate the influence of both node strength and number of connections over the data being parsed [2,21]. See weighted formulas below. Betweenness was calculated in Gephi software as unweighted scores.

In-degree Centrality Calculation:Weighted In-degree = K_i_^(1−α)^ × S_i_^α^(1)
where, K_i_ = number of in-degree connections; S_i_ = in-degree weight; α = 0.5 or 1.0

Out-degree Centrality Calculation:Weighted Out-degree = K_o_^(1−α)^ × S_o_^α^(2)
where, K_o_ = number of out-degree connections; S_o_ = out-degree weight; α = 0.5 or 1.0

Closeness Calculations:Closeness = [∑ S_T_^α^] ^−1^(3)
where, S_T_ = total degree weight between two nodes; α = 0.5 or 1.0

Social networks are made up of social members and their relationships, where as little as 10 top-ranking individuals can significantly influence a whole network; these individuals make up a small portion of the population and are referred to as super-spreaders [11]. In the present study, we are more interested in potential super-spreaders rather than other nodes within the network. Since ranking is used to predict potential super-spreaders [11] and we had close to 100 male WTD in our study (*n* = 96), we compared only the top 20 highest-ranking potential super-spreaders for out-degree centrality, in-degree centrality, and betweenness centrality scores by age group (young males vs. mature males). The age groups were compared based on average scores and their prevalence rate within the top 20 highest-ranking potential super-spreaders. Statistical analysis was performed using a one-way ANOVA test, where *p* < 0.05 was considered significant. All distributions were tested using the Jarque–Bera test [19] with a threshold of *p* = 0.05 and appeared to be normally distributed. We hypothesized that predatory activity (combination of hunters and predators) would decrease the scraping activity of mature males at the different study site locations. To test the effect of predatory activity on scraping activity, we first measured the predatory activity at each major study site. Next, we examined the scraping activity of young and mature males at each major study site. Scraping activity and predatory activity were compared across the Major Study Sites 1, 2, and 3 using the ANOVA test, where *p* < 0.05 was considered significant. We then compared Mature and Young scraping activity vs. predatory activity using a correlation analysis by plotting scraping activity against predator activity for both mature and young males across Major Study Sites 1, 2, and 3. We also surveyed the number of hunters and male WTD harvested in the surrounding areas near the major study sites (Table 2). To determine the impact of hunter-harvest on scraping networks, a Hunter Harvest Regional Network was generated by removing the nodes of male WTD harvested (removed WTD, *n* = 21) from the Regional Network (*n* = 96). The Regional Network data were compared to the Hunter Harvest network using network statistics with relevance to disease transmission (Table 1) using an ANOVA test, where *p* < 0.05 was considered significant. For assessing management strategy models, we removed the top 20 out-degree, in-degree, and betweenness super-spreaders from the Regional Network and compared network statistics with relevance to disease transmission (Table 1) between network models using an ANOVA test, where *p* < 0.05 was considered significant. Data sets were not tested for normal distribution due to *n* values ranging from 96 to 75 and assumed to be normally distributed based upon the central limit theorem [25]. To model indirect transmission risk, where scrapes are potential sources of infection, we generated a scrape-to-scrape network using scrapes as nodes and the unique males common to scrape pairs as non-directional weighted edges. Graphs and network statistics such as betweenness and weighted degree were generated in Gephi software.

### 2.4. QGIS Mapping and Hotspot Prediction

Scrape sites were mapped using QGIS software as previously described [2]. Scrapes are potential indirect contacts of disease transmission [14], where some scrapes may pose a higher threat as disease transmission hotspots. Transmission hotspots are location clusters, where the magnitude of indirect or direct contacts between infected and uninfected individuals are greater than other locations, thus increasing the risk for disease transmission at these locations [26]. We examined many scrape characteristics such as number of unique bucks, scrape activity, network density, scrape betweenness, scrape weighted degree, and hotspot formulas to locate potential disease transmission hotspots. We suggest that transmission hotspots are influenced by all of these scrape characteristics. Therefore, a hotspot is comprised of a significant number of unique males visiting the scrape, considerable scrape activity, and a highly connected local community having a strong network density. Therefore, we compared the product of these scraping characteristics for predicting transmission hotspots using two formulas.

Hotspot Formula:Hotspot = (number of unique bucks) × (scrape activity) × (network density of the local community)(4)

Alternative Hotspot Formula:Hotspot = (weighted degree of a scrape) × (network density of the local community)(5)

Our first hotspot formula is calculated using data from a social network with known scraping activity data. The alternative formula combines data from a scrape-to-scrape network with data from a social network. These values and scrape characteristics were statistically compared using a Z-test. A Z-test can be used to determine patterns and differences within spatial data [15,27], where Z-scores are expressed in standard deviation units from mean of the population. A Z-score <−1.65 or >+1.65 is significant at a 90% confidence level and a Z-score <−1.96 or >+1.96 is significant at a 95% confidence level [15,27]. We define a transmission hotspot as an area with a cluster of scrapes with values that are significantly higher than the population mean of all scrapes. The QGIS heatmapping function was used to visualize hotspots based on the number of unique males, scraping activity, network density, betweenness, weighted degree, or hotspot formulas at each study site location, scaled from blue to red. For predicting community crossroads, betweenness centrality scores of each scrape site generated from the scrape-to-scrape network were compared using a Z-test at a 90% confidence level. Betweenness scores were used to create a heatmap over Sites 1, 2, and 3 to locate community crossroads using QGIS 3.28.0.

## 3. Results

### 3.1. Generating Multiple Community WTD Scraping Networks

We captured over 118,000 digital images and identified 96 unique male WTD, where on average 7.3 ± 5.5 unique males visited each scrape (Table 2). Using the scrape location and scraping activity of each unique WTD male, we generated social networks for each major study site and calculated the most dominate male (Figure 2A–C). We combined the social networks for Site 1, Site 2, Site 3, and the boundary sites (not shown) to generate the Regional Network (Figure 2D). Community analysis predicted four communities within the Regional Network stemming from Site 1, Site 2, Site 3, and Boundary 3.

### 3.2. Scraping Networks Depict Direct Social Contacts

Mean closeness scores of 29 bachelor pairs were compared using the Regional Network and the Random Network (Figure 3A–C). The Regional Network generated significantly lower closeness scores as compared to the Random Network suggesting the Regional Network is a superior predictor of WTD bachelor groups. Using the same Regional and Random Networks, we compared the mean closeness scores for male WTD seen sparring or fighting (Figure 3D). Once again, the Regional Network generated significantly lower closeness scores as compared to the Random Network suggesting the Regional Network is a superior predictor of physical altercations between male WTD. Overall, these data support closeness scores from scraping networks as indicators of direct contact.

### 3.3. Identifying Potential Super-Spreaders Using Scraping Networks

In-degree, out-degree, and betweenness centrality scores for each male from the Regional Network were used to identify the highest-ranking potential super-spreaders in each category. The highest-ranking males from each super-spreader category were used to demonstrate transmission risk across the Regional Network (Figure 4A). F38 was the top-ranking out-degree potential super-spreader capable of infecting 31 direct social contacts (Figure 4B). H10 was the top-ranking in-degree potential super-spreader capable of infecting 42 direct social contacts (Figure 4C). F09 was the top-ranking betweenness potential super-spreader capable of infecting 61 direct social contacts (Figure 4D). We examined the top 20 potential super-spreaders for each category. There were 36 males total in the top 20 potential super-spreaders for each category, because some of the same males were in multiple categories. For example, F09 was in the top 20 ranking for out-degree, in-degree, and betweenness potential super-spreaders. We examined the age structure of the top 20 highest-ranking males based upon out-degree centrality, in-degree centrality, and betweenness centrality scores to determine if age correlated with potential super-spreader scores. Mean out-degree and in-degree centrality scores were not significantly different among young males and mature males, meaning that both young and mature males are equally capable of spreading disease as out-degree and in-degree potential super-spreaders (Figure 5A,B). The combined out-degree and in-degree centrality scores for both young and mature males were found to be normally distributed. However, the combined betweenness centrality scores for both young and mature males were found not to be normally distributed, indicating a large set of outliers. The mean betweenness centrality scores were significantly higher for young males as compared to mature males, suggesting that young males were outliers and were more capable of spreading disease as betweenness potential super-spreaders as compared to mature males (Figure 5C). Next, we examined the prevalence rate of young males or mature males in the demographics of the top 20 potential super-spreaders. Interestingly, young males outnumbered mature males and had a higher prevalence rate in the top 20 out-degree, in-degree, and betweenness potential super-spreaders, representing up to 93% of the top 20 potential super-spreaders (Figure 5D). All the top-ranking betweenness potential super-spreaders were young males (Figure 5E). Therefore, as a disease-management strategy, reducing the number of young male WTD within the population would decrease the potential spread of communicable diseases within this Regional Network.

### 3.4. Predatory Activity and Hunting Activity Influences Scraping Networks

According to the current literature, most scraping behavior is performed by mature males >2.5 years old occurring near the peak of the WTD breeding season [2,5,18], where mature males scrape to suppress younger males from breeding [1,5,8]. At Site 1, there was no significant difference between young and mature male scrape activity (Figure 6A). At Site 2, mature males scraped significantly more often as compared to young males (Figure 6A). However, young males scraped significantly more often as compared to mature males at Site 3 (Figure 6A). This may be due to the fact that Site 3 had significantly more predator activity/hunting activity as compared to Sites 1 and 2 (Figure 6B) with up to eight predators/hunters present every week (>1/day). Correlation analysis revealed an increase in young male scraping activity with increased predator activity and a decrease in mature scraping activity with increased predatory activity (Figure 6C,D). Site 3 averaged 4.3 hunters per 100 acres compared to 0.6 and 2.8 at Site 1 and Site 2, respectively. A total of 23 male WTD from the Regional Network were harvested by hunters (Table 2), where 11 male WTD were harvested at Site 3 as compared to 3 total WTD harvested at Site 1 and Site 2 combined (Table 2). These data suggest locations with higher predatory activity/hunting activity may reduce mature male scraping activity or presence at those locations, where the lack of mature male scraping presence may increase young male scraping activity influencing the demographics of local networks.

### 3.5. Hunter Harvest and Potential Super-Spreader Management Reduces Transmission Risk

Social network disease management models were generated from the Regional Network by removing hunter harvested WTD or by removing the top 20 highest-ranking out-degree, in-degree, and betweenness potential super-spreaders (Figure 7). Using network measures, we compared the Regional Network before (*n* = 96) and after hunter harvest (*n* = 75) and found a significant reduction in transmission risk and outbreak potential (Table 3). Hunter harvests significantly reduced the average degree, weighted degree, triangles per node, and path length (*p* < 0.05, Table 3). The network density of the Regional Network was reduced by almost half post-hunter harvest suggesting hunter harvests reduced outbreak potential. Eliminating the top 20 highest-ranking out-degree, in-degree, and betweenness potential super-spreaders also significantly reduced disease transmission risk and lowered outbreak potential (Table 3). Removing out-degree, in-degree, and betweenness potential super-spreaders significantly reduced the average degree, weighted degree, triangles per node, and path length (*p* < 0.05, Table 3). Network density was also reduced by nearly half. Interestingly, removal of out-degree or betweenness potential super-spreaders increased the number of communities within the regional network suggesting that removal of these types of super-spreaders lowers disease transmission risk and outbreak potential by fragmenting the network into less connected communities (Table 3 and Figure 7). Removal of out-degree super-spreaders had the most impact on disease transmission risk which significantly reduced the average degree, weighted degree, triangles per node, and path length values as compared to all other network models (Table 3). Hunters harvested 50% (18 out of 36) of the top-ranking potential super-spreaders: 6 out-degree, 8 in-degree, and 4 betweenness potential super-spreaders (Table 4). Interestingly, after polling the hunting community, we found that youth hunters harvested as many potential super-spreaders as adult hunters, where a single youth hunter harvested the highest ranking betweenness potential super-spreader F09; F09 was also a high-ranking out-degree and in-degree super-spreader (Table 4; Figure 5E). Adult hunters indicated that they mostly targeted mature males for harvest and youth hunters were allowed to harvest both young and mature males. A total of 80% of the male deer harvested by youth hunters were young males, and 38% of the male deer harvested by adult hunters were young males. Therefore, management strategies, such as controlled hunting of young males, aids the removal of out-degree and betweenness super-spreaders and greatly reduces disease transmission and outbreak potential. We suggest that population management, such as regulated hunting, focused on elimination of young males as opposed to trophy hunting (mature males) would help to maintain or improve the health status of WTD populations. 

### 3.6. Potential Transmission Hotspots and Community Crossroads

Scrape characteristics such as the number of unique bucks and scrape activity were used to predict transmission hotspots within major study Sites 1, 2, and 3. The number of unique bucks and scrape activity displayed similar results, where only one significantly different scrape was found (Figure 8 and Appendix A). We suggest that transmission hotspots are influenced by all scrape characteristics and the social connectivity of the local community. We derived a new formula to locate hotspots using scrape characteristics, network density, and Z-scores and predicted a hotspot of three significantly different scrapes clustered together at Site 3 (Figure 8A and Appendix A). Since scrapes are also plausible indirect routes of transmission, we analyzed scrapes within the Regional Network to create the first scrape-to-scrape network, where scrapes are nodes and unique bucks common to scrapes served as edges (Figure 8B). To better visualize indirect transmission potential, we examined the weighted degree and betweenness scores of scrapes to determine scrape sites that are critical for spreading disease. Scrape weighted degree Z-scores revealed two significantly different scrapes spread far apart, but no hotspot (Figure 8A and Appendix A). We derived an alternative formula to locate hotspots using the scrape weighted degree, the network density of the local community, and Z-scores and found the same hotspot of three significantly different scrapes clustered together at Site 3 as predicted with our previous hotspot formula (Figure 8A and Appendix A). Using scrape betweenness scores and Z-scores, we located four significant scrapes serving as potential disease transmission crossroads between communities (Figure 8C and Appendix A). Overall, we demonstrate that scraping characteristics and social networks influence the risk of both direct and indirect disease transmission, where these factors can be used to predict the location of transmission hotspots and community crossroads.

## 4. Discussion

Understanding communicable disease transmission among WTD populations has become a major public health concern, as WTD can be infected with the zoonotic virus, SARS-CoV-2 [12,28]. Chronic wasting disease (CWD), a highly contagious prion disease, is another disease of concern for WTD and other cervids [13]. Social network analysis has provided communicable disease modeling, which has proven useful for developing control strategies for human contagions such as HIV, the SARS virus, SARS-CoV-2, and even *E. coli* outbreaks to name just a few [29,30,31,32,33]. Network analysis has also provided insight for management strategies for livestock diseases [34,35,36]. In this study, we demonstrate that WTD social-network models can be used to identify potential super-spreaders within WTD communities. We also demonstrated how WTD social networks can be used to locate potential disease transmission hotspots and community crossroads. This modeling information can then be used to develop more effective disease-management strategies. These strategies are vitally important at our research location in Yazoo County, Mississippi, which borders three CWD positive counties [37]. 

A recent study using GPS modeling concluded that direct contact drives CWD transmission in free-ranging WTD [38]. Further, local patterns of CWD infection are highly influenced by social structure and social interactions [39]. Social disease transmission is directed by super-spreaders, which are infected individuals that infect more social contacts than other infectious individuals within a population due to their social rank within social networks [9,10,11,26]. Identifying potential super-spreaders is a key component used to develop disease-management strategies and pandemic-preparedness strategies [40]. Similar to previous animal social network studies, we found that out-degree and betweenness centrality were important network measures for assessing an individual’s ability to spread an infection [41]. We also looked at the prevalence rate of highest-scoring potential super-spreaders and found that young males significantly outnumbered mature males, further suggesting that young males are key players in disease transmission within this WTD network (Figure 5). Early modeling studies concluded that these centrality measures are also important predictors of an individual’s risk of infection as well as a predictor for the time to infection during a disease outbreak [42]. Thus, high-scoring potential super-spreaders would have a greater infection risk and become infected early on during a disease outbreak. Studies in WTD CWD and SARS-CoV-2 infections support our findings. Early on in the Illinois CWD outbreak, prevalence was higher for younger males (1.5–2.5 years old) and lower for older males (4.5–5.5+ years old) [43]. As CWD progressed over many years in Illinois, studies showed higher rates in adult deer as compared to young WTD [44], which can be explained by CWD’s long incubation period in an aging population [45]. A recent serosurveillance study of SARS-CoV-2 infections in Texas found significantly higher percentages of positives in younger WTD as compared to older WTD [46]. 

The social behavior and home range size of these younger male WTD is most likely the cause of the higher infection rates as compared to older males. Younger males are reported to have larger home ranges as compared to mature males [47]. We suspect that the young male betweenness centrality originates from male dispersal. The dispersal of young males is caused by a combination of social cues, mate competition, male–male altercations, and inbreeding avoidance between the dam and her male offspring [48,49,50]. Therefore, dispersing males cover greater distances, form more social contacts, are at higher risk for infections, and become potential super-spreaders of disease [51]. So, as young males disperse, they are more likely to come into contact with prions in a CWD positive area. This may be one explanation as to why CWD prevalence is two times higher in male than in female WTD [38,52,53,54,55].

Studies have suggested that a reduction in male WTD populations would provide an effective means of CWD management by reducing both the prevalence and frequency of infection [56,57]. A 2011 study, focused on WTD movement, found that movements of young males posed the greatest threat for rapid disease transmission from infected populations and that disease management should focus on reducing young male populations [51]. Our data agree with these management strategies and suggest that targeting younger males, early on and throughout the hunting season, would further reduce the frequency of infection and lower disease prevalence in cases such as CWD or SARS-CoV-2 outbreaks.

Other studies have shown that hunting pressure reduces WTD movement, which further supports hunting as a disease-management strategy [58]. The impact of hunter harvests on our regional network data further supports this management strategy. Hunter harvests significantly decreased the number and weight of male WTD social contacts, lowered the overall network density by fragmenting the network, and therefore reduced the risks of disease transmission and lowered outbreak potential (Table 3). Removal of the top 20 out-degree and betweenness super-spreaders reduced the risk of transmission, lowered outbreak potential, and fragmented the network into more communities (Figure 7 and Table 3). Animal social network studies have shown that fragmented social networks structurally trap infections within a few subgroups delaying the spread of disease outbreaks [41,59]. Community fragmentation was demonstrated plainly with the removal of the top-ranking betweenness super-spreaders (Figure 7E). Youth hunters played a key role in this network fragmentation, where a single youth hunter harvested the highest ranking betweenness super-spreader, demonstrating the importance of teaching the next generation of WTD hunters (Table 4).

We also found that hunter activity and predator activity influenced the age structure of males scraping in WTD communities (Figure 6). In communities with less hunter and predator activity, mature males scraped more often or equal to that of younger males. However, in areas with higher hunter and predator activity, younger males scraped more often that mature males, having a major impact on the demographics of the local network (Figure 2). This is no surprise as hunting activity has been shown to influence WTD movement both spatially and temporally to avoid potential contact with hunters [58,60]. Furthermore, recent studies show that WTD display behavioral changes during non-human predation [61,62]. These findings support further investigations to determine the influence of hunting and predator activity on WTD social networks and disease transmission networks. Overall, we found that young male WTD are the most threatening potential super-spreaders and posed the highest risk for disease acquisition and transmission; therefore, increasing harvests of young male WTD would be a logical disease prevention and management strategy at this location.

Another driver of CWD infection in WTD, and other deer species, is transmission through indirect or surface contact with infectious particles within the environment. Infectious prions can linger in the environment for years, thus increasing in concentration as disease prevalence grows [16,63,64,65]. A comprehensive study on WTD scraping behavior suggests that scrape sites are most likely a reservoir for CWD prions and a source for indirect disease transmission [14]. Recent studies indicate that prions are shed from multiple WTD scent glands used in scraping behavior [13]. Both prions and SARS-CoV-2 have been detected in the nasal and oral secretions of asymptomatic and symptomatic WTD further supporting scrapes as potential indirect sources of infections [14,28,66].

In this study, we expanded on this idea using QGIS and network data to locate possible transmission hotspots and community crossroads (Figure 8). Potential transmission hotspots were determined using our new hotspot formula combining the number of unique bucks scraping, scrape activity, and network density of the local community (Figure 8). Not all scrapes are created equal and vary by scrape activity and by the individual WTD working the scrape [2,3,14]. The highly active and popular scrapes have been called “Community Scrapes” and are the most likely sources of local, indirect disease transmission [1,67]. As a disease-management strategy, these scrapes can be removed from use by destroying the licking branch, which is considered to be the highest risk component of the scrape [14]. To better understand indirect disease transmission between WTD communities, we developed the first scrape-to-scrape network and used betweenness centrality to locate community crossroads (Figure 8). Community crossroads are a new concept derived from this study and represent connected locations, such as scrapes, where WTD communities overlap, and the risks of indirect infections are high. Our model suggests that once an infectious particle reaches one of these community crossroads, the infection will then spread into neighboring communities. Management strategies targeted at transmission hotspots would reduce disease spread within communities and management of community crossroads would reduce transmission between communities.

Taken together, our modeling data warrants further research into the social aspects of direct and indirect disease transmission among WTD communities, especially in locations where communities overlap. Finding infectious particles, such as CWD prions or SARS-CoV-2, on licking branches or in scrape soils would further support these transmission models. Overall, this study sheds light on the importance of WTD social networks in transmission risk models providing new insight for WTD disease prevention and outbreak management strategies.

## Figures and Tables

**Figure 1 animals-13-01171-f001:**
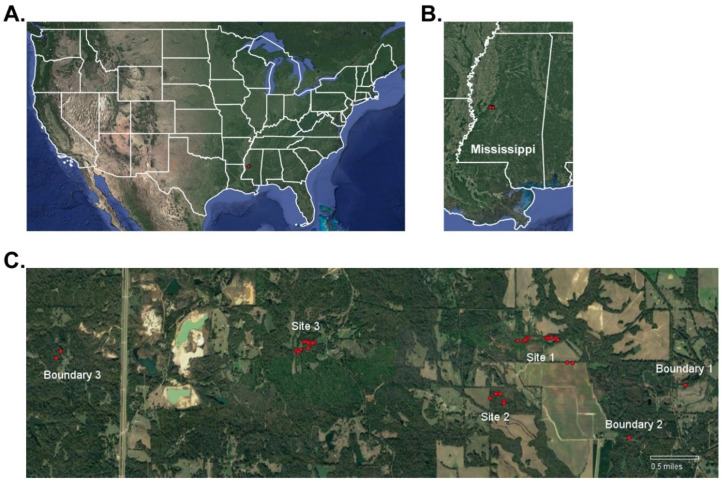
Study location in Bentonia, Mississippi, USA. Shown is the study location (**A**) at the national level and (**B**) at the state level marked by a red circle. (**C**) Shown is a zoomed map of the study location, which is comprised of 3 major study sites and 3 boundary sites. Scrape sites monitored by camera traps are shown as red circles throughout the study location. Scale bar: 0.5 miles.

**Figure 2 animals-13-01171-f002:**
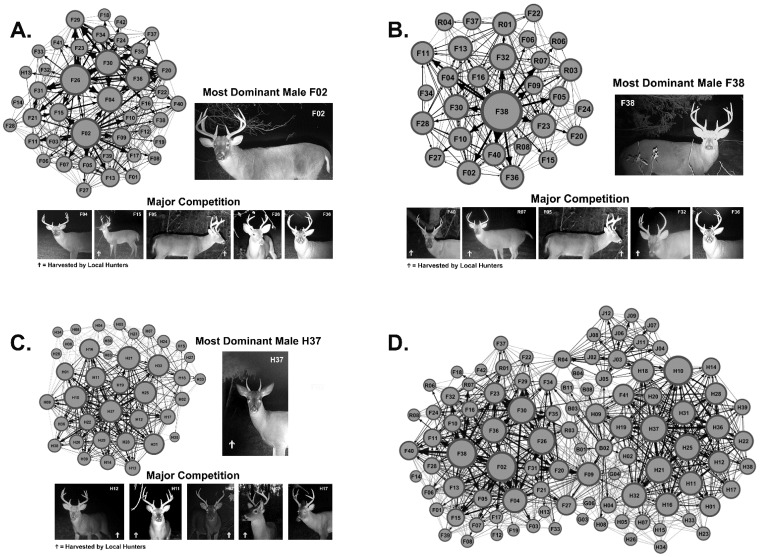
Local and Regional Social Networks. Shown are the local community networks and the region-wide social network generated using scraping network analysis. (**A**) Shown is the local community network generated from Site 1 data and images of the most dominate male F02 and his major competition. The node size in each graph varies as a factor of total weighted degree. Ϯ Indicates male WTD harvested by local hunters. (**B**) Shown is the local community network at Site 2 and images of the most dominate male F38 and his major competition. (**C**) Shown is the local community network at Site 3 and images of the most dominate male H37 and his major competition. (**D**) Shown is the region-wide social network (i.e., Regional Network) generated from Sites 1, 2, and 3 as well as boundary sites’ data.

**Figure 3 animals-13-01171-f003:**
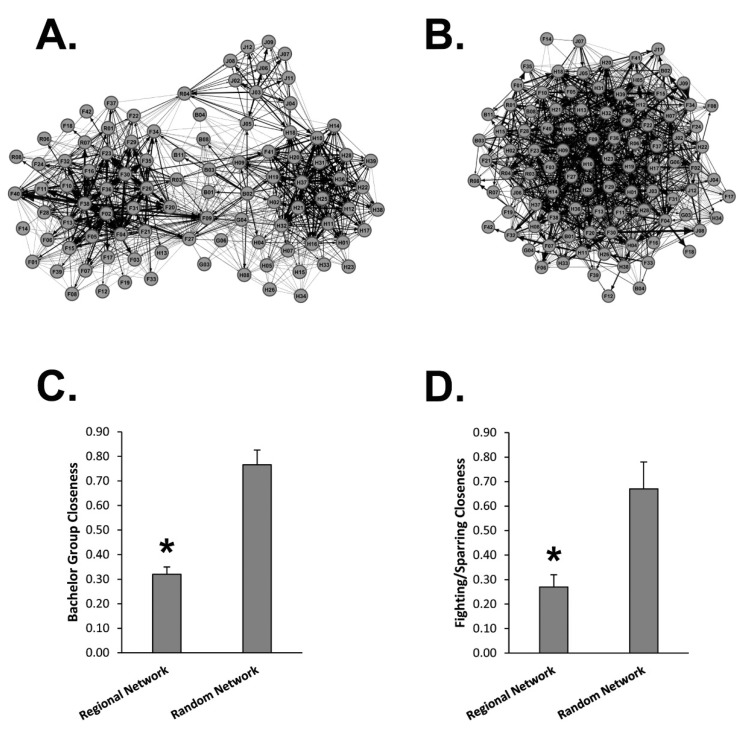
Regional Network vs. Random Network. (**A**) Shown is the Regional Network graph derived from scraping data at Site 1, Site 2, Site 3 and all boundary locations. (**B**) Shown is the Random Network graph derived from randomization of the Regional Network data. (**C**) The graph displays the Avg ± Stdev of closeness scores between bachelor pairs measured using the Regional Network and the Random Network (*n* = 29). (**D**) The graph displays the Avg ± Stdev of closeness scores between sparring or fighting pairs measured using the Regional Network and the Random Network (*n* = 9). Statistical analysis was performed using a one-way ANOVA test, where * *p* < 0.05 was considered significant.

**Figure 4 animals-13-01171-f004:**
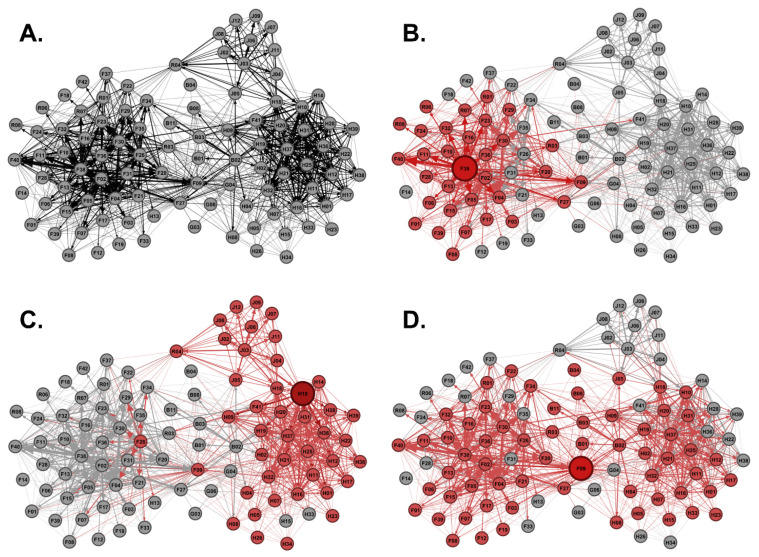
Modeling Potential Super-Spreaders. (**A**) Shown is the Regional Network generated from Major Sites 1, 2, and 3 as well as boundary sites’ scraping data. (**B**) Shown is the disease transmission risk of the highest-ranking out-degree potential super-spreader within the Regional Network, where the infected node F38 is enlarged and colored red forming 31 direct contacts. Contact nodes and weighted edges are also colored red to emphasize risk. (**C**) Shown is the disease transmission risk of the highest-ranking in-degree potential super-spreader, where the infected node H10 forms 42 direct contacts. (**D**) Shown is the disease transmission risk of the highest-ranking betweenness potential super-spreader, where the infected node F09 forms 61 direct contacts.

**Figure 5 animals-13-01171-f005:**
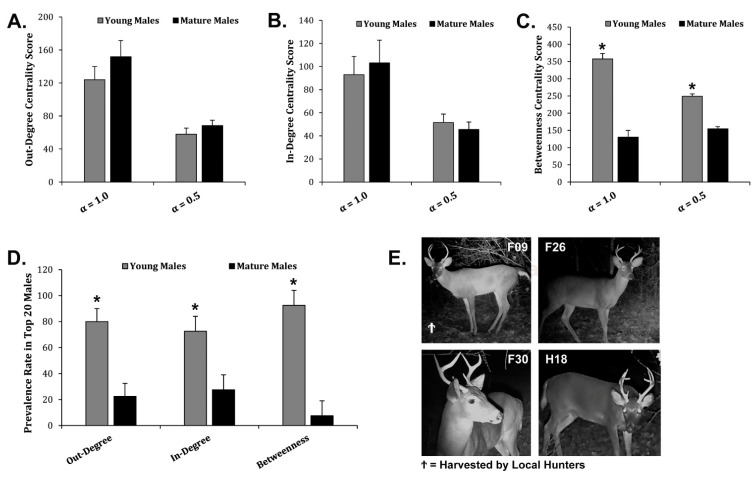
Age Structure of Potential Super-Spreaders. (**A**) The graph displays the Avg ± Stdev of out-degree centrality scores comparing young males and mature males using α = 1.0 or α = 0.5. (**B**) The graph shows the Avg ± Stdev of in-degree centrality scores comparing young males and mature males using α = 1.0 or α = 0.5. (**C**) The graph displays the Avg ± Stdev of betweenness centrality scores comparing young males and mature males using α = 1.0 or α = 0.5. (**D**) The graph shows the Avg ± Stdev of the prevalence in the top 20 males that were either young males or mature males among the top 20 out-degree, in-degree, and betweenness highest-ranking males. Statistical analysis was performed using a one-way ANOVA test, where * *p* < 0.05 was considered significant. (**E**) Shown are digital images of the highest ranking betweenness potential super-spreaders.

**Figure 6 animals-13-01171-f006:**
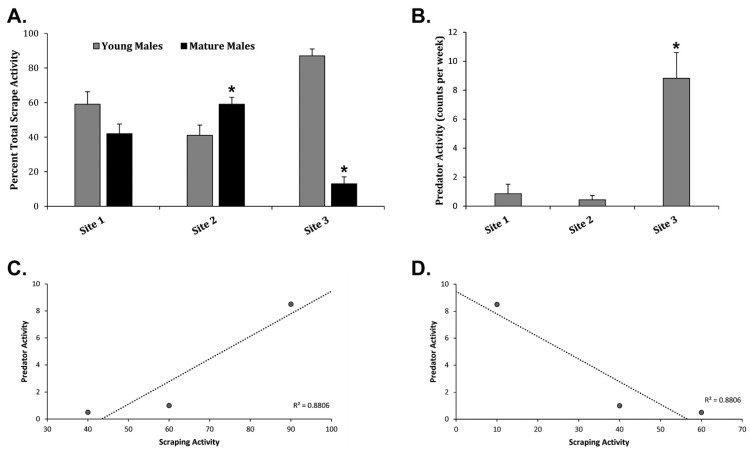
Influence of Predation on Scraping Activity. (**A**) The graph shows the Avg ± Stdev of the percentage of total scrape activity among young males and mature males at Sites 1, 2, and 3. (**B**) The graph displays the Avg ± Stdev of predator activity in counts per week at Sites 1, 2, and 3. Statistical analysis was performed using a one-way ANOVA test, where * *p* < 0.05 was considered significant. (**C**) The graph displays young male scraping activity plotted against predator activity with a correlation coefficient of 0.88. (**D**) The graph displays mature male scraping activity plotted against predator activity with a correlation coefficient of 0.88.

**Figure 7 animals-13-01171-f007:**
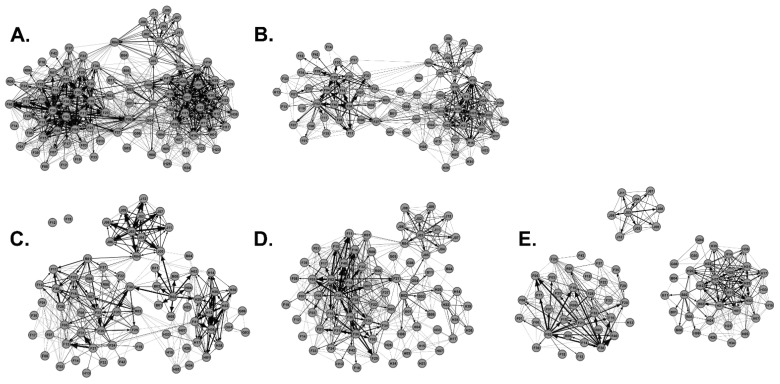
Impact of Hunter Harvest and Super-Spreader Management on Social Networks. Shown is (**A**) the Regional Network, (**B**) the Regional Network post-hunter harvest, (**C**) the Regional Network minus the top 20 out-degree potential super-spreaders, (**D**) the Regional Network minus the top 20 in-degree potential super-spreaders, and (**E**) the Regional Network minus the top 20 betweenness potential super-spreaders.

**Figure 8 animals-13-01171-f008:**
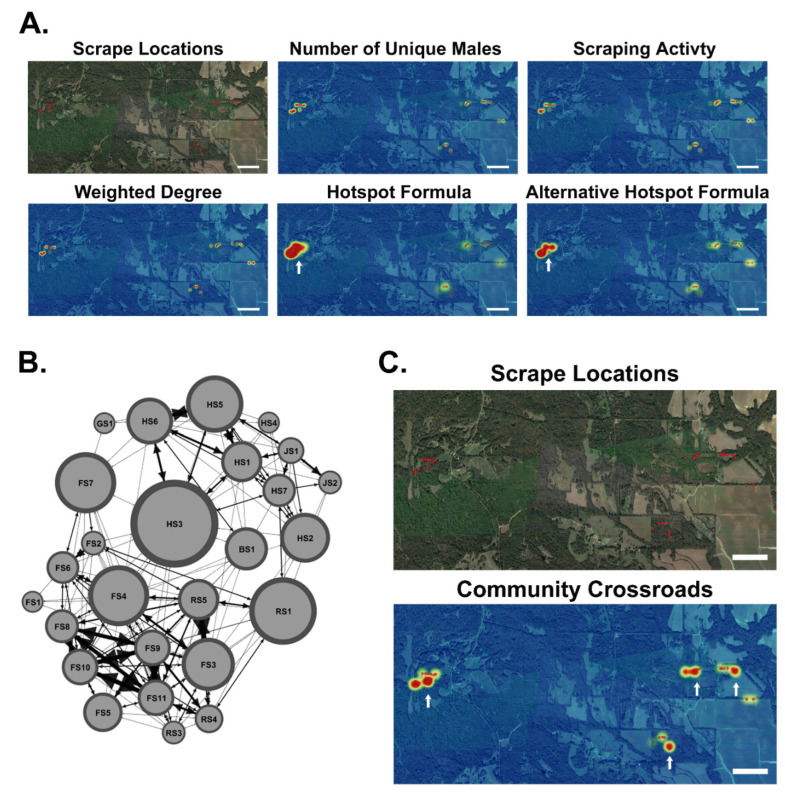
Disease Transmission Hotspots and Community Crossroads. (**A**) Shown is the base map of the scrape locations within Major Sites 1, 2, and 3. Scale bar: 0.5 miles. Shown are overlaid heatmaps of the number of unique males scraping at each scrape, scraping activity at each scrape, scrape weighted degree, and transmission hotspots located using our hotspot formula and our alternative hotspot formula. White arrows point to transmission hotspots (outliers found using the Z-scores). Scale bar: 0.5 miles. (**B**) Shown is the scrape-to-scrape network graph generated using scrapes as nodes and common males as edges. The node size reflects betweenness centrality scores. (**C**) Shown is a map of the scrape locations within Sites 1, 2, and 3. Shown is a heatmap of betweenness centrality scores highlighting community crossroads. White arrows point to community crossroads. Scale Bar: 0.5 miles.

**Table 1 animals-13-01171-t001:** Network Measures and Disease Relevance. The table displays the definition of each network measure and its relevance in disease modeling.

Social Network Measure	Disease Relevance
**Degree:** the total number of incoming and outgoing connections an individual has in a network.	Individuals with high degree are more likely to become infected and spread infection during an outbreak.
**Weighted Degree:** the frequency of an individual’s social interactions with other individuals within a network.	Higher degree weights correlate with increased risk of disease transmission within social groups.
**Outdegree:** the number of outgoing connections an individual has in a network (can be weighted or unweighted).	Individuals with high outdegree scores have a greater potential to spread disease to more individuals within a network.
**Indegree:** the number of incoming connections an individual has in a network (can be weighted or unweighted).	Individuals with high indegree scores are at a greater risk of becoming infected from multiple individuals within a network.
**Betweenness:** the number of times an individual occurs on the shortest path between two other individuals within the network.	Betweenness scores describe the potential of an individual to spread infection by bridging multiple individuals or communities within a network.
**Closeness:** the path length from an individual to another individual in the network.	Smaller closeness values indicate a closer relationship between individuals, where disease transmission via direct contact is more likely to occur.
**Triangles:** the number of groups of three connected individuals.	The more triangles within a network represent strong connectivity and a higher potential for disease outbreak as compared to less connected networks.
**Average Path Length:** the average social distance between individuals within a network.	Networks with smaller average path transmit disease more efficiently as compared to networks with larger average path lengths.
**Network Density:** the proportion of connected individuals out of all possible connections within a network.	Network density scores measure the disease outbreak potential within a network. Networks with higher network density are at a greater risk of disease outbreak as compared to networks with lower network density.
**Community:** a group of nodes within a network that have a higher probability of being connected to each other as compared to the rest of the network.	A network with a small number of highly connected communities has a greater outbreak risk as compared to networks with a larger number of less connected communities.
**Transmission Hotspot:** a cluster of location points with characteristics that are significantly higher than the population mean.	Hotspots are locations where the disease transmission risk or outbreak potential is higher as compared to other locations.
**Community Crossroads:** areas where multiple communities overlap or intersect.	Locations with high betweenness scores form community crossroads, where disease transmission from one community to another is most likely to occur.

**Table 2 animals-13-01171-t002:** Site Specific Data and Network Statistics. The table shows data collected at Site 1, Site 2, Site 3, and over the entire Regional Network. Shown are the number of images taken, number of scrapes surveyed, number of unique males at each site, Avg ± Stdev of unique males per scrape, min to max of unique males per scrape, ratio of young to mature males, hunters per 100 acers, number of male WTD harvested, and Avg ± Stdev of predator activity (counts per week). Shown are the network statistics calculated for Site 1, Site 2, Site 3, and over the entire regional network. Shown are the weighted and unweighted connections, and network density.

Location:	Site 1	Site 2	Site 3	Regional Network
Digital Images Taken	49,600	20,344	31,577	118,195
Number of Scrapes	13	6	10	33
Number of Unique Males	42	29	39	96
Unique Males per Scrape (Avg ± Stdev)	5.2 ± 3.0	5.0 ± 3.5	12.3 ± 7.1	7.3 ± 5.5
Unique Males per Scrape (Min to Max)	1 to 12	1 to 10	3 to 25	1 to 25
Ratio of Young Males to Mature Males	2.9	2.2	4.4	3.2
Unweighted Connections	476	318	1170	1729
Weighted Connections	2494	1600	4162	9368
Network Density	0.28	0.39	0.88	0.22
Hunters per 100 Acers	0.6	2.8	4.3	2.6
Male Deer Harvested	3	0	11	23
Predator Activiy (Avg ± Stdev)	0.9 ± 0.6	0.4 ± 0.3	8.8 ± 1.8	3.4 ± 4.7

**Table 3 animals-13-01171-t003:** Impact of Hunter Harvest and Disease Management on Network Statistics. The table shows the changes in the Regional Network statistics before and post-hunter harvest, and after removal of the top 20 out-degree, in-degree, and betweenness super-spreaders. Changes in Avg degree, Avg weighted degree, Avg triangles per node, Avg path length, network density, and number of communities were compared before and post-hunter harvest, and after removal of out-degree, in-degree, and betweenness super-spreaders. Statistical analysis was performed using a one-way ANOVA test, where *p* < 0.05 was considered significant.

Network Model	Average Degree	Average Weighted Degree	Average Triangles per Node	Average Path Length	Network Density	Number of Communities	Significance to Regional Network	Significance to All Networks
Regional Network (*n* = 96)	35.9	98.2	172.6	1.98	0.22	4	n/a	n/a
Hunter Harvested Network (*n* = 75)	24.4	74.8	74.8	2.19	0.16	4	*p* < 0.05	Not Significant
Out-degree Spreaders Removed (*n* = 76)	16.9	33.9	40.5	2.58	0.11	6	*p* < 0.05	*p* < 0.05
In-degree Spreaders Removed (*n* = 76)	19.6	50.1	65.1	2.39	0.13	4	*p* < 0.05	Not Significant
Betweenness Spreaders Removed (*n* = 76)	18.5	48.3	51.8	1.68	0.12	5	*p* < 0.05	Not Significant

**Table 4 animals-13-01171-t004:** Hunter Harvested Potential Super-Spreaders. The table displays the number of high-ranking out-degree, in-degree, and betweenness potential super-spreaders within the region-wide network harvested by hunters. Also shown is the number of potential super-spreaders harvested by youth and adult hunters.

Potential Super-Spreaders	Deer Harvested	Youth Hunter Harvested	Adult Hunter Harvested
Out-Degree	6	4	2
In-Degree	8	4	4
Betweenness	4	1	3
Total	18	9	9

## Data Availability

The data presented in this study are available upon request from the corresponding author. The data are not publicly available due to the large storage requirements for digital images.

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
