# Peer review of "Identifying Potential Super-Spreaders and Disease Transmission Hotspots Using White-Tailed Deer Scraping Networks"

_animals, 2023, doi:10.3390/ani13071171_

Round 1

Reviewer 1 Report (Previous Reviewer 3)

I have reviewed the new version of the paper entitled “Identifying Potential Super-Spreaders and Disease Transmission Hotspots Using White-tailed Deer Scraping Networks” submitted by Hearst, S. and colleagues. I would like to congratulate the authors for the effort the have made to improve the manuscript.

Here you have some extra considerations to improve the manuscript:

-          I would consider changing the title of the manuscript in order to include all the parameters considered in the present study since you also include analysis regarding predation and hunting

-          Regarding one of my previous considerations, I cannot see why sometimes analysis includes information from all the study points (major study sites and boundaries), and in others the boundaries are removed from the analysis (for example: for predatory activity, table 2, hotspot prediction…).

-          In order to test the effect of predatory activity, you could consider checking for differences between the scraping activity of the same male in different points and then try to test the effect of predation, since differences between scraping activity in the different sites could answer to other explanation (type of vegetation, local density of animals…).

-          It would be interesting to see the predation activity in the different sites (table2)

-           Figure 6: you could consider representing plots C and D together, since both represent almost the same

-          Figure 8A (Scrape locations): the points cannot be seeing, consider modifying the map using more contrast in the points

-          Line 400: I would specify that the management strategies, such as controlled hunting, would reduce potential disease transmission. Since the paper works with potential scenarios, is necessary to be careful with the statements.

-          Line 398: it would interesting to see the age distribution of hunted bucks between youth and adult hunters.

-          Line 395: not “just as many”, youth hunters hunted exactly the same super-spreaders as adult hunters according to table 4. Besides, the case of F09 hunted by a youth hunter is irrelevant in my opinion, since it has no replica or statistical value, so it could be a random fact. This fact brings me to my next point:

-          The recommendations that you give throughout the entire manuscript in relation to the health management of the populations should give advice applicable at a global or more general scale, not to a local scale. Therefore, from my point of view, the key idea that should be extracted from this work is that hunting is a useful tool in the sanitary management of populations, since hunting management focused on the elimination of young males against hunting of trophies (adult males) will help maintain or improve the health status of populations. So the age of the hunters is totally irrelevant, and therefore I think this should be removed from the paper.

Author Response

Reviewer 2 Report (New Reviewer)

“Identifying potential super-spreaders and disease transmission hotspots using white-tailed deer scraping networks”

The authors used network analysis to establish the significance of White-tailed Deer social networks in transmission of diseases. The study is of importance in predicting super-spreaders of diseases and location of transmission hotspots.

The manuscript of generally well-written, but some grammatical errors throughout the paper need to be corrected.

Introduction

Background information and justification for the study is provided in the Introduction but different concepts should be presented as paragraphs, rather than one continuous script.

L81. “locate” instead of “located”

L92 “identify” instead of “identified”

Methods

Figure 1. The map is showing the study locations, but these locations should also be shown on State- and National-level maps.

L101. Add the country after “Mississippi”

L115. Delete the first “IRB”

L142. “defined” instead of “define”

L150: Rephrase as “Statistical analysis was performed using………”

L151: “considered” instead of “consider”. Same with L195, L201 and elsewhere

L161: “data” instead of “statistics”

Discussion

L459: SARS-COV-2 is not a disease, but the virus that causes the disease.

L461: “control” not “controls”. Similar grammar errors in line 466

Author Response

Reviewer 3 Report (New Reviewer)

At first sight, the study seems flawless, as it is well described and presented. However...

My major concern regards to the lack of knowledge on infectiology and epidemiology of infectious diseases. The authors equate the behaviour of a species with the certainty of transmitting pathogens causing diseases as diverse as CWD and SARS-CoV-2. In order to reach the final conclusion that certain animals are super-spreaders more evidence is needed. On the basis of presented analysis what we can say, that those animals are ‘the super contacts’, not that they are responsible for virus/prion transmission. It is not proved by any epidemiological data or disease modelling in WTD.

The study is based on male-based transmission theory, however it has not been proven in any studies on the epidemiology and transmission of those two diseases that transmission was sex-dependant. Was it? Further, deer-to-deer transmission is not as efficient as spill-over from humans for SARS-CoV-2.

As for CWD, the indirect transmission mode has been overlooked completely. However, it plays and important role in disease epidemiology.

The use of references which are cited in the Discussion (lines 487-492) to confirm the observations is questionable. They show perhaps age-dependent exposure (prevalence/seroprevalence), however not ability of young males to spread the pathogens and become super-spreaders.

A statement: ‘CWD prions have been detected in multiple WTD scent glands used in scraping behavior’ suggesting that sniffing directly affects prion protein to amplify in lymphoid tissue is. This shows authors lack of knowledge on the pathology and epidemiology of diseases they write about.

I would rather suggest that the authors concentrate on the ecology of the species, space use and behaviour, social networking, rather than analyse pathogen transmission model.

Nevertheless, the research and analysis of the results is impressive and worthy of publication.

If the authors insist on keeping the study – transmission model study, I suggest, for an article to be accepted for publication, thorough analysis and correction by an epidemiology expert (preferably in CWD and/or SARS-Cov-2) is recommended first.

Author Response

This manuscript is a resubmission of an earlier submission. The following is a list of the peer review reports and author responses from that submission.

Round 1

Reviewer 1 Report

This manuscript presents fascinating data on the risk of disease spread through scraping networks by white-tailed detailed. The authors have compiled an impressive dataset from 33 scrapes and their findings appear to have important implications. Unfortunately, I found the paper lacking in important details, poorly organized, and replete with typographical errors. The Results section includes as much description of methods as it does actual results. The Discussion section was just two very long and rambling paragraphs (one paragraph being nearly a page and a half). The paper could potentially be a valuable contribution if the writing was made more concise and organization improved. Detailed comments follow.

1.                   Line 67: According to Alexy et al, this should be greater than or equal to (including 2.5-year olds).

2.                   Lines 77-79: Alexy et al reported that the same individual rarely visited multiple scrapes.

3.                   Line 92: Why report only the width and not the length? To have this area, it must have been less than a half-mile long. Also, the longer axis is typically referred to as the length, i.e., 6 miles long x 0.5 miles wide. Also, unless journal format calls for English units, convert miles to km and don't report acres.

4.                   Line 93-94: Please provide more detail, e.g., dominant species composition, crops farmed, etc.

5.                   Line 103: Please explain “3/5 mode.” How frequently were images taken? Any delay? Was there a burst of images for each trigger event? What was the likelihood that you missed behaviors between images?

6.                   Line 103: How were scrapes located? Systematic searches? Randomly walking the area? Please describe.

7.                   Line 112: What are the study sites and boundary sites? They haven't been described.

8.                   Line 120: Please describe sites 1, 2, and 3 in the study area section which only describes one area. This is the first mention of 3 sites.

9.                   Line 122: I'm not sure a t-test is appropriate. It sounds like you are comparing 3 means and a t-test is for comparing 2 means. Why not an ANOVA?

10.               Line 148: “were compared statistically.” Statements like this are made throughout and are not informative. How were they compared statistically? A reader should be able to reproduce what you did from your description.

11.               Line 160: “using methods as previously described.” Also used throughout and also not informative. Don't make the reader go look up another paper to discover your methods. It is fine to cite another paper as the basis for a method but it still needs to be described at least in summary, with reference to the other paper for additional detail.

12.               Line 170-172: This sentence is not Results. It belongs either in the Introduction where objectives are described or in the Discussion.

13.               Line 172-174: Move to Study Area.

14.               Line 178: “the most dominant male.” As defined how? By the most visits? If so, dominance does not seem an appropriate term, especially considering the "dominant" male in Fig 2C is a yearling and others pictured from that scrape are obviously mature.

15.               Lines 209-213: Move to Methods.

16.               Lines 229-232: Move to Methods.

17.               Line 277: Panel E is not described.

18.               Lines 279-286: This includes both methods and results. Separate the methods and describe in that section.

19.               Lines 294-295: Since this was based on surveying hunters, how confident are you that you accounted for all males that were harvested, i.e., was your survey a complete census of hunters and their harvest?

20.               Lines 324-332: Methods.

21.               Lines 335-347: Methods.

22.               Lines 375-377: Sentence fragment.

23.               Line 395: Do you mean “serosurveillance”?

24.               Line 491: Miller, K. V.: The “V” initial is lowercase throughout the Lit Cited. Correct to uppercase.

25.               References: several citations are missing page numbers. Please check formatting throughout.

Reviewer 2 Report

The authors report on the use of social network analysis to detect possible super-spreaders among a White-Tail Deer Population. This approach is quite innovative and can put the base for further research on the ecology of wildlife diseases. I would like to thank the authors for this very well written manuscript.
Several issues should be addressed before the manuscript can be considered for publication. I hope that the following observations and suggestions may help the authors improving this manuscript.

Specific comments:

18 Since it isn't demonstrate that this is the case I would use the term "possible disease transmission hotspots".

20 Add "and hunting" after "predators"

27 Odocoileus virginianus has to be written in italic front.

37 Actually not only direct contact but also indirect contacts. I would indicate both.

47 In the text you just consider "predator activity" including also the hunters. See 118-119. Please clarify this point.

How were the study sites and the boundary sites selected? Please add additional information regarding this. How can we be sure that there aren’t other relevant area between the sites? This can be even be more relevant than those selected for the studies? Those can change the results of the super-spreader deer?

96-97 I am not familiar with the hunting system in the USA. Can be possible that hunters not living close to the survey location would hunt animals there? If yes, would they be a small portion of the hunters hunting there? How can you exclude that you haven't underestimate the hunting pressure in these locations?

103 I would also precise that the scapes were selected during a previous study

106-107 Please precise if you examined images or videos? It would be quite difficult to recognize the described behaviours only with pictures. The use of images would underestimated the detection of such behaviour since image represent one single moment during a longer frame of movmenets. How did you define an "event"? Did each scraping behaviour of the same animal counted as 1 event if occured within a certain time frame? Or how did you do?
Did you take a screenshot of the videos to create the catalog (111)?

119 How did you define a "feral dog"? Due to the presence or the abscence of a collar?

122 Student's T-test: did you test if your values were normally distributed?

138 Why the comparison between Real Network and Null network was done only for Site 1? I would suggest the authors to perform the analysis also for sites 2 and 3.

L141-142: "Bachelor grouping behaviour" and "sparing/fighting behaviour" should be defined.

L147: How did you decide to consider the first 20 animals? Why not more? Why not less?

149-150: see also 96-97. The Post-Hunter Harvest regional network would be underestimated if not all hunters were included in the survey. How can you be sure that you did not miss any hunted animals?

157-158: see also 122. did you test if your values were normally distributed?

163-164: how and when did you define "scraping activity" ?

Table 1. Number of Scrapes?

Description of Table 1: A lot of repetition. Not necessary.

Results

The results section has several portions that should actually go tot he M&M (e.g. 209-214, 324-336; including the hypotheses: 283-286, 297-298) and tot he discussion  (e.g. 266-268), respectively.

191-192: What do you mean with "Number of Scrapes"? Is this the number of scrapes event seen during the entire study period in the different sites?

Fig. 2/3: Figure 2A and 3A reppresent the Real Network of Site 1. Why do you reppresent it in two different ways? It would be less confusing it would be the same image (with of course a similar size oft he nodes)

224. Please consider to write "Average + Standard deviation" rather than "Avg +Stdev" thorought the manuscript

227. Statistical significance should been used between two measurements. In this case closeness score between Real network and null network. I would suggest to put the * on a line above both barst to indicate the the difference between the two is statistical significant.
Alternatively, the sentence "where * p > 0.05 was considered significant" should be reformulated.

Please consider these changements thorought the entire manuscript.

235. "capable of infecting". I would stay more vague and say "potentially capable of infect"

Fig. 5E description of the image is missing. Is it necessary?

277. See 227. The sentence "where * p > 0.05 was considered significant" should be reformulated.

Figure 6. Both Y-axes should have a similar format. In the legend of Y-Axe of Figure 6A  "Percentage" should be put in parentheses at the end, similarly to 6B. Especially in Fig 6B has to be put more clearly for which bar there is a significant difference between the bars. As for 122 and 157-158, did you test for normality oft he values? Is the T-test an appropriate test for your values?

294: Not knowing the expected population size of WTD in each site the absolute number of hunted deer is not very indicative. It would be more helpful to have a percentage of deer hunted in a specific site.

307-308: The age of hunter doesn't say a lot regarding their experience with hunting. It would be much more important to differentiate between year of hunting experience. At least should be indicated how you defined young and adult hunters.

318: see also 122 and 157-158 Did you test if your values were normally distributed?

Fig. 7a. Consider insert a scale bar in each image.

Discussion

371 Consider insert a "potential" before "disease transmission hotspots"

391 Please indicate Illinois instead of «IL» for not USA-readers

415-417/427-429 Can you discuss here the effect of hunting on the increased movement of animals and ist impact on disease transmission? What is their role in spreading the disease?

It would be worth to discuss more in detail what are the implication for management to detect hotsport for transmission. What can you practically do when hotspots are idfentified? Can the management of this locations be improved in ordert o increase / avoid transmission? How?

Reviewer 3 Report

I have reviewed the paper entitled “Identifying Potential Super-Spreaders and Disease Transmission Hotspots Using White-tailed Deer Scraping Networks” submitted by Hearst, S. and colleagues. The paper evaluates the use of social network analyses to detect potential super-spreader and hotspots of transmission within a population of white-tailed deer (Odocoileus virginianus).

Although the work shows extensive results and several interesting points, I have found several points that make me consider the adequacy of this work to be published, especially in relation to the methodology used and the novelty of the work.

Major comments

My major concern is regarding the methodology employed to describe the nodes of the network performed to detect potential super-spreader. The authors state that they cataloged individuals using camera traps according to the methodology presented by Jacobson et al. 1997 and Hearst, S. et al. 2021., however, this same author in his study of 2021 states that individual recognition using digital images has shown to be less accurate in rural settings (which seems to be the study area) that in suburban areas. I agree with this statement since I consider it difficult to identify individuals of species that do not have a unique pattern or another trait identifiable using camera traps. Therefore, I find that the network could be vague if we cannot correctly identify the nodes and the edges of it.

The authors aim to study potential super-spreaders and transmission hotspots using as model diseases CWD and SARS-CoV-2, however, it would be much more accurate considering the biology and ecology of these diseases in the creation of the two networks presented by including the surviving time of both pathogens in the environment as a temporal window to consider an edge between individuals.

 In general, the present manuscript does not seem to add great novelty to the previous work (Hearst, S. 2021) apart from the scrape-to-scrape network and the identification of super-spreader using the network statistics provided by gephi. Besides, social network analysis in epidemiology has been long used. The authors should therefore state better during the introduction and discussion the novelty of the work and the contributions to the field.

In general I see many descriptive results, but the statistics are vague and focused only on data provided by gephi program. The authors could consider modelling the results recorded and extrapolating them to the area around the sites according to census, vegetation, hunting activity….

Specific comments

1.       Introduction

- I would include in this section why scrapes would imply transmission of the two pathogens presented

- Please also include the role of harvesting or predators (since you comment these factors in the abstract) and why it is interesting to include these parameters in the network

- It is necessary to better state the novelty of the work regarding the previous study and the contribution to the field of the present study

2.       Methodology

In general, the methodology is confusing and vague, and it is necessary to reach the results section to understand the methodology employed by the authors. Authors need to explain much more in detail the methodology employed even if it is the same as the methodology employed in previous works. Here you have some examples:

-  You need to reference an image of the study area

- Which are the points included in the study, which are the differences between site and boundaries

- Explain in detail how to get the percentage of scrape

- Line 93: EEUU

- Line 100: Which means “male WTD” cataloged”

- Line 102: How do you find and describe scrapes?

- Line 110: explain according to what parameters you cataloge males

- It is necessary to include more in detail information about the study area: type of vegetation, hunting statistics, predator census, WTD census, age ratio of hunting deers….

-  Line 128: weighted edges according to what?

- Line 128-129: do you consider any time window between consecutive visits to scrape points? It would be interesting to consider the surviving time of the pathogens of interest as time window

- Line 133: it would be interesting to add a table (maybe as supplementary material) of the network statistics explained in detail and how if affects the aim of the study

- Line 138: why site 1? What happened with site 2, 3 and the boundaries?

-   Line 163: why density?

-   Line 163-164: maybe it could also be interesting to include degree

3.       Results

-  Line 170-185: consider transferring this to methodology

- Figure 1: it needs to be referenced in methodology. Besides, in methodology you need to explain the conditions of each of the studied points (site 1, 2, 3 and boundaries)

-  Line 211: why with site 1 and not with the regional network?

- Line 212-220: consider transferring this to methodology because you don’t understand why it is necessary to use a null network up to this point

-  Figure 3A: this is a node-to-node network or scrape-to-scrape?

- Line 230-232: consider transferring this to methodology

- Figure 4: consider transferring this figure to supplementary material

-  Line 266-268: consider transferring this to discussion

- Figure 5: you need to explain what alpha means (also in methodology section)

- Line 290-293: consider including some of this information in the methodology section.

- Line 290: predator activity seems to be irrelevant since you don’t use this information later

- Line 302: average triangles?

- Line 306: age of hunters seems to be irrelevant to the goal of the study. Besides, the results presented don’t show apparently significant differences between the age of the hunters and the deer harvested.

- Table 2: which are the differences between total and weighted degrees? And triangles?

- Line 324: consider transferring this to introduction

-  Figure 7: what about boundaries? What is the role of boundaries? (you seems to exclude this information in all the statistics)

- Line 335: transfer to discussion

-  Line 339: what happened with site 2 and 3?

-  Line 347: I don’t understand the link between betweenness scores and bridge points

- Line 348-351: transfer this to conclusion or discussion

- Figure 7: these images do not provide much information. Consider restructuring the information, removing the images o providing a zoom of the areas studied

4.       Discussion

-  Line 377-378: this information is already present in the manuscript

- Line 391: better add the state before the capital letter the first time

-  Line 420: I don’t trust this statement since results do not seem to be significantly different between hunters. Also, along the study you state that the super-spreaders are related to betweenness parameter, and young hunters hunted fewer deer with high betweenness values. I think that management recommendation should be that hunting activity needs to be regulated by disease management (both young and adult males) instead of hunting focused on trophy purposes (adult males)

- Line 438-441: consider transferring this to introduction of justification

-  Line 452: which model suggests this? Please explain in detail

- I miss a more extensive paragraph detailing the contribution of the work, also some conclusions